# Using Macrophage Polarization in Human Platelet Lysate to Test the Immunomodulatory Potential of Cells for Clinical Use

**DOI:** 10.3390/biomedicines12040833

**Published:** 2024-04-09

**Authors:** Silvia Lopa, Francesca Libonati, Katia Mareschi, Giuseppe Talò, Stefania Brambilla, Vincenzo Raffo, Luciana Labanca, Luigi Zagra, Matteo Moretti, Laura de Girolamo, Alessandra Colombini

**Affiliations:** 1Cell and Tissue Engineering Laboratory, IRCCS Istituto Ortopedico Galeazzi, Via C. Belgioioso 173, 20157 Milan, Italy; silvia.lopa@grupposandonato.it (S.L.); giuseppe.talo@grupposandonato.it (G.T.); stefania.brambilla@grupposandonato.it (S.B.); 2Orthopaedic Biotechnology Laboratory, IRCCS Istituto Ortopedico Galeazzi, Via C. Belgioioso 173, 20157 Milan, Italy; libonati.francesca@gmail.com (F.L.); vincenzo.raffo@grupposandonato.it (V.R.); alessandra.colombini@grupposandonato.it (A.C.); 3Department of Public Health and Paediatrics, University of Turin, Via Verdi 8, 10124 Turin, Italy; katia.mareschi@unito.it; 4Stem Cell Transplantation and Cellular Therapy Laboratory, Paediatric Onco-Haematology Division, Regina Margherita Children’s Hospital, City of Health and Science of Turin, 10126 Turin, Italy; 5Blood Component Production and Validation Center, City of Health and Science of Turin, S. Anna Hospital, 10126 Turin, Italy; llabanca@cittadellasalute.to.it; 6Hip Department, IRCCS Istituto Ortopedico Galeazzi, Via C. Belgioioso 173, 20157 Milan, Italy; luigi.zagra@fastwebnet.it; 7Regenerative Medicine Technologies Laboratory, Laboratories for Translational Research (LRT), Ente Ospedaliero Cantonale (EOC), Via F. Chiesa 5, CH-6500 Bellinzona, Switzerland; 8Service of Orthopaedics and Traumatology, Department of Surgery, EOC, Via Tesserete 46, CH-6900 Lugano, Switzerland; 9Euler Institute, Faculty of Biomedical Sciences, Università della Svizzera Italiana (USI), Via la Santa 1, CH-6962 Lugano, Switzerland

**Keywords:** macrophage, human platelet lysate, polarization, chondrocyte, co-culture

## Abstract

Macrophage-based co-cultures are used to test the immunomodulatory function of candidate cells for clinical use. This study aimed to characterize a macrophage polarization model using human platelet lysate (hPL) as a GMP-compliant alternative to Fetal Bovine Serum (FBS). Primary human monocytes were differentiated into unpolarized (M0) or polarized (M1, M2a, and M2c) macrophages in an hPL- or FBS-based medium. The protein secretion profiles and expression of phenotypic markers (CD80 for M1, CD206 for M2a, and CD163 for M2c) were analyzed. Subsequently, chondrocytes were tested in an hPL-based co-culture model to assess their immunomodulatory function in view of their possible use in patients with osteoarthritis. The results showed similar marker regulation between hPL and FBS cultures, but lower basal levels of CD206 and CD163 in hPL-cultured macrophages. Functional co-culture experiments with chondrocytes revealed increased CD206 expression both in hPL and in FBS, indicating an interaction between macrophages and chondrocytes. While markers in FBS-cultured macrophages were confirmed in hPL-cultured cells, the interpretation of marker modulation in immunomodulatory assays with hPL-based cultures should be carried out cautiously due to the observed differences in the basal marker levels for CD206 and CD163. This research underscores the utility of hPL as a GMP-compliant alternative to FBS for macrophage-based co-cultures and highlights the importance of understanding marker expressions in different culture conditions.

## 1. Introduction

Osteoarthritis (OA) has long been regarded as a cartilage disease, but increasing evidence implies that low-grade inflammation is a hallmark of this pathology. Inflammation is mainly mediated by macrophages infiltrated in the synovium [1,2] and is one of the main factors responsible for the initiation and chronicization of the degenerative mechanisms leading to joint destruction. In the joint, macrophages are induced by the local microenvironment to exhibit different phenotypes and polarization states, with recent evidence showing that M1 macrophages are elevated in both synovium and circulation during OA development, along with lower numbers of M2 macrophages [3]. Pro-inflammatory M1 and anti-inflammatory M2 macrophages represent the two extremes of a plethora of macrophage differentiation states that can be found in vivo [4]. M2 macrophages can be further divided into different subpopulations, which include M2a macrophages, which are mainly related to anti-inflammatory activity, and M2c macrophages, which are mainly related to tissue repair [5].

In this context, establishing a functional co-culture model is crucial to evaluate the ability to modulate the macrophage polarization of candidate cells for the treatment of inflammation in patients with OA [6,7,8,9,10,11]. Co-culture models have been widely applied in OA research, mimicking joint microenvironments to study complex cell interactions [12]. These models can be exploited to elucidate cell–cell communication, unraveling OA pathogenesis and identifying potential therapeutic targets, and they include different combinations of articular cells and/or tissue explants, offering insights into OA’s complexity [13,14,15]. Co-culture systems, including macrophages, are instead more focused on understanding OA-related inflammation and testing the ability of cell-based therapies to modulate the inflammatory state [10,16]. To this aim, assessing markers of the macrophage phenotype is a reliable method to determine if candidate cells can induce macrophage switch from a pro-inflammatory to an anti-inflammatory phenotype.

Co-culture models are usually established in standard fetal bovine serum (FBS)-based culture. Despite indisputable advantages related to the use of FBS, there are several quality and safety concerns that depend on its poor characterization, heterogeneous composition, immunogenicity, and infection-related risks [17]. For this reason, in recent decades, alternative ancillary materials of human origin have been tested to support cell growth, especially for the preparation of clinical-grade cell-based products. Among them, human platelet lysate (hPL) represents a valuable option as a non-xenogenic Good Manufacturing Practice (GMP)-compliant supplement for culture media when cells require an expansion phase before being delivered to patients [18,19,20,21]. hPL is a cell-free supernatant that is rich in growth factors, such as platelet-derived growth factors, transforming growth factor beta, insulin-like growth factor-1, brain-derived neurotrophic factor, vascular endothelial growth factor, epidermal growth factor, basic fibroblast growth factor, hepatocyte growth factor, connective tissue growth factor, and bone morphogenetic protein-2, -4, and -6 [22], which are released after the freeze–thawing degranulation of platelet concentrate (PC) prepared by apheresis as a standard blood product in transfusion medicine. Currently, the thorough characterization of cells cultured in hPL intended for clinical use, especially for the treatment of inflamed microenvironments, should also include the analysis of their immunomodulatory behavior. This is possible by assessing the ability of candidate cells to modulate the phenotype and function of relevant immune cells in specific co-culture models in the presence of hPL if this supplement is used for cell expansion prior to clinical cell administration.

The possibility to culture, differentiate, and polarize macrophages in the presence of hPL is an almost unexplored field. To the best of our knowledge, the literature reports only one study by Tylek and colleagues in which hPL was used for macrophage culture and polarization experiments [23], demonstrating that using hPL is a suitable option to cultivate and differentiate primary human macrophages. Additionally, another recent study has investigated the effect of adding hPL to FBS-cultured M1 macrophages [24], proving that it has an anti-inflammatory effect.

With macrophages being highly plastic cells that promptly respond to soluble mediators, it is of great importance to establish a standard method for testing the immunomodulatory properties in hPL-supplemented culture medium and to verify whether the phenotypic markers normally used in standard FBS-based culture conditions can still be considered as reliable test results.

Based on these premises, the aim of this study was to generate a macrophage polarization model in hPL, assessing the secretory profiles and the expressions of CD80, CD206, and CD163, which are, respectively, considered as M1, M2a, and M2c markers [25,26,27,28,29]. Primary human macrophages cultured and polarized in hPL or FBS were compared to assess if the specific phenotypes shared the same typical markers in the two different culture conditions. The experimental validity of the model was then verified in a functional assay of macrophage and chondrocyte co-culture to mimic the crosstalk between these cell types in a joint microenvironment in view of the use of chondrocytes as cell-based therapy in patients with OA.

## 2. Materials and Methods

### 2.1. Experimental Design

A schematic representation of the experimental aims and corresponding methodology is illustrated in Figure 1. The timeline and analyses applied for macrophage differentiation and polarization in hPL or FBS is depicted in Figure 1a. Primary human monocytes were differentiated for 5 days in the presence of M-CSF in either hPL- or FBS-based culture medium. Afterwards, macrophages were maintained unpolarized (M0) or induced towards M1, M2a, and M2c phenotypes by cytokine-based stimulation for 2 days. On day 7, macrophage phenotype was analyzed by protein array, gene expression, and flow cytometry analysis.

The timeline and methodology applied for the co-culture set-up are presented in Figure 1b. Primary human monocytes were cultured for 7 days in the presence of M-CSF to obtain M0 macrophages. On day 5, human chondrocytes were seeded in transwell and cultured in hPL-based culture medium. On day 7, the co-culture phase was initiated. Macrophages were co-cultured with chondrocytes for 2 days. At the end of the co-culture, macrophages were analyzed by flow cytometry analysis.

### 2.2. Preparation of hPL

The hPL was prepared from a platelet pool of 60 healthy donors at the Blood Component Production and Validation Center, City of Health and Science of Turin, S. Anna Hospital. Briefly, 450 mL of whole blood was collected using a quadruple bag system (Fresenius Kabi, Bad Homburg vor der Höhe, Germany) containing citrate-phosphate-dextrose. The donors were tested for ABO blood groups, irregular red blood cell antibodies, and infectious markers (hepatitis B and C, human immunodeficiency virus 1/2, and Treponema pallidum). After centrifugation, the blood units were separated into plasma, buffy coat, and red blood cells using an automated blood component separator (Compomat G5, Fresenius Kabi). Buffy coat platelet concentrates (BC-PC) were prepared by pooling 5 O-group buffy coats with one AB-group plasma unit and processing them with the TACSI system (Terumo BCT Europe, Leuven, Belgio) to achieve a platelet concentration of approximately 1000 × 10^6^ platelets/µL, determined by Sysmex XE-2100 (Sysmex Corporation, Kōbe, Japan).

The BC-PCs were ultimately inactivated using the Mirasol PRT System (Terumo BCT Europe). To induce platelet fragmentation and growth factor release, the BC-PCs underwent three cycles of freezing (−35 °C) and thawing (37 °C). To remove platelet bodies, the BC-PCs were finally centrifuged (5000× *g*, 8 min), and the supernatant was collected. To improve standardization and reduce individual donor variations, ten supernatant units were pooled into a single HPL unit, where 200 IU/mL of heparin were added. The hPL unit was then divided into aliquots of 100 mL and frozen again at −35 °C until use after thawing.

### 2.3. Isolation of Monocytes

Human primary white blood cells were isolated by Ficoll (GE Healthcare, Chicago, IL, USA) density gradient separation from 16 buffy coats of healthy donors obtained from the local blood bank. Monocytes were then obtained by positive magnetic selection using CD14 microbeads (MACS, Miltenyi Biotec, Bergisch-Gladbach, Germany). After isolation, monocytes were counted and frozen in liquid nitrogen using hPL or heat-inactivated FBS with 10% (*v*/*v*) DMSO added at a concentration of 10 × 10^6^ cells/vial.

### 2.4. Macrophage Differentiation and Polarization

After thawing, monocytes obtained from multiple donors were pooled together to constitute three different batches of cells. Pooling of monocytes from multiple donors was needed to achieve a suitable number of cells for the subsequent experiments.

For each pool, 40 × 10^6^ cells were seeded at a density of 4 × 10^5^ cells/cm^2^ in RPMI 1640 (Sigma-Aldrich, St. Louis, MO, USA) supplemented with 10% (*v*/*v*) hPL and 100 U/mL penicillin, 100 µg/mL streptomycin, and 200 mM glutamine (ThermoFisher Scientific, Waltham, MA, USA). The other 40 × 10^6^ cells were seeded in the same culture condition supplemented with 10% (*v*/*v*) heat-inactivated FBS. To differentiate monocytes into macrophages, 20 ng/mL of macrophage colony-stimulating factor (M-CSF, Peprotech Inc, Rocky Hill, NJ, USA) was added to hPL- and FBS-supplemented medium [25,30,31].

After 2 days, non-adherent cells were re-plated, and medium was refreshed. After 3 more days, macrophages were polarized into M1, M2a, and M2c for 2 days, following different protocols: M1 phenotype was induced with IFN-γ (100 ng/mL, Peprotech, Cranbury, NJ, USA) and LPS (100 ng/mL, Sigma-Aldrich), the M2a phenotype was induced with IL-4 (40 ng/mL, Peprotech) and IL-13 (20 ng/mL, Peprotech), and the M2c phenotype was induced with IL-10 (40 ng/mL, Peprotech) [25,30,31]. All of the polarizing media contained 20 ng/mL M-CSF. Macrophages cultured only with M-CSF until day 7 represented the unstimulated macrophages (M0).

### 2.5. Chondrocyte Isolation and Expansion

A pool of chondrocytes was obtained from 5 different donors subjected to total hip arthroplasty. The study protocol (MS-TIP) for the collection of primary human chondrocytes from patients’ biopsies was approved by the San Raffaele Hospital Ethics Committee on 16 December 2020, registered under number 214/int/2020. The articular cartilage obtained from superficial areas of the femoral head was cut into small pieces and enzymatically digested with 0.15% (*w*/*v*) Type II Collagenase (Worthington Biochemical Corporation) for 22 h at 37 °C [32]. Cells were seeded at a density of 1 × 10^4^ cells/cm^2^ in High Glucose Dulbecco’s modified Eagle’s medium (DMEM) (Sigma-Aldrich) supplemented with 10% (*v*/*v*) hPL, 10 mM HEPES, 1 mM sodium pyruvate, 200 mM L-Glutamine, 100 U/mL penicillin, and 100 µg/mL streptomycin, namely chondrocyte medium (all from Thermofisher Scientific), and incubated at 37 °C. After 7 days, when confluence was reached, the cells were detached, further expanded at 5 × 10^3^ cells/cm^2^ in expansion medium up to three passages (21 days), and frozen for the subsequent experiments.

### 2.6. Co-Culture Model

Pooled monocytes were thawed and seeded in 6-well plates (2 × 10^5^ cells/cm^2^) and cultured for 7 days in macrophage culture medium (RPMI 1640 medium with addition of 20 ng/mL M-CSF) supplemented with either hPL or FBS to obtain M0 macrophages. On day 5, a chondrocyte pool was thawed, seeded in transwell (1.5 × 10^4^ cells/cm^2^), and cultured in DMEM-based culture medium supplemented with either hPL or FBS. After 2 days (day 7), the transwells were transferred to the macrophage-seeded 6-well plates to initiate the co-culture phase. Macrophages were co-cultured with chondrocytes for 48 h using a mix of RPMI-based and DMEM-based medium (1:1). At the end of the co-culture, macrophages were detached, stained for cell surface markers, and analyzed by flow cytometry, as already described.

### 2.7. Image Acquisition and Analysis

Cells cultured in hPL- or FBS-based medium were imaged on day 2, day 5, and day 7 with a phase contrast microscope (Olympus IX71, Hamburg, Germany). The images were analyzed using the image processing software Fiji (ImageJ 1.54f). A macro was programmed to set a manual threshold to identify all of the particles in the image. The selected particles were converted to a mask and finally analyzed with the command “Analyze Particles”. All particles above 30 µm^2^ were included. The particle area and shape descriptors were extracted and saved in an .xls file for statistical analysis.

### 2.8. Multiplex Immunofluorescence Assay for Cytokine/Chemokine Quantification

Cell supernatants were collected from M0, M1, M2a, and M2c macrophages cultured either in hPL or FBS on day 7. The supernatants were centrifuged at 500g for 5 min and then stored at −80 °C. A multiplex bead-based immunofluorescent assay (R&D Human Magnetic Luminex Customized Assay, R&D Systems Inc., Minneapolis, MN, USA) was used to determine the protein secretion profiles of the macrophages. For the analysis, cell supernatants were thawed and incubated with a cocktail of antibodies pre-coated onto magnetic microparticles. A cocktail of biotinylated antibodies was added, followed by streptavidin–phycoerythrin conjugate. Finally, the microparticles were read using a MagPix™ System (Bio-Rad Laboratories, Inc., Hercules, CA, USA). Samples were analyzed in duplicate. A 5-parametric standard curve (Bio-PlexManager software, version 6.1.1, Biorad) was used to determine the protein concentration of samples. For each analyte, data were subtracted from background values measured in hPL-based or FBS-based medium and maintained at 37 °C for 2 days in the absence of cells. Additionally, the residual values of IL4, IL13, and IL10 added to the medium at the same concentrations used for M2a and M2c polarization and kept at 37 °C for 2 days were subtracted from the values measured for these cytokines in the supernatants collected from the M2a and M2c macrophages.

### 2.9. Gene Expression Analysis

On day 7, unpolarized and polarized macrophages cultured in hPL- or FBS-based medium cells were lysed using the lysis buffer provided in the PureLink^®^ RNA Mini Kit (ThermoFisher Scientific, Waltham, MA, USA). RNA was isolated from cell lysates according to the manufacturer’s instructions, quantified spectrophotometrically (NanoDrop, Thermo Scientific), and reverse-transcribed to cDNA by employing the iScript cDNA Synthesis Kit (Bio-Rad Laboratories). Gene expression was evaluated by real-time PCR (StepOne Plus, Life Technologies). Briefly, 13 ng cDNA was incubated with a PCR mixture, including TaqMan^®^ Gene Expression Master Mix and TaqMan^®^ Gene Expression Assays (Life Technologies). The following assays were used: CD80 (Hs01045162_m1) for M1 phenotype; CD206 (Hs00267207_m1) for M2a phenotype; and CD163 (Hs00174705_m1) for M2c phenotype.

GAPDH (Hs99999905_m1), UBC (Hs00824723_m1), and RPL13A (Hs04194366_g1) were tested as potential housekeeping genes, and RPL13A was selected for its stability. Data were normalized on the housekeeping gene (2^−ΔCt^ method).

### 2.10. Flow Cytometry Analysis of Surface Markers

To analyze the macrophage immunophenotype, a flow cytometry analysis was conducted on day 7 on unpolarized (M0) and polarized macrophages (M1, M2a, and M2c) cultured either in hPL- or FBS-based medium. The cells were washed twice with phosphate-buffered saline (PBS), detached by incubation with cell dissociation buffer (Thermo Fisher) for 7 min, and centrifuged at 500g for 5 min. Macrophages were then suspended in MACS buffer (Miltenyi Biotec), treated with FcR Blocking Reagent (Miltenyi Biotec) for 10 min at 4 °C, and counted. Afterwards, 10^5^ cells were stained to evaluate the expression of cell surface markers with the following antibodies: anti-human CD80-APC (Clone REA661, Miltenyi Biotec) for M1 phenotype, anti-human CD206-FITC (Clone 15-2, Biolegend, San Diego, CA, USA) for M2a phenotype, and anti-human CD163-PE (Clone GHI/61, Biolegend) for M2c phenotype. Unstained cells were used as negative controls for fluorescence. The appropriate isotype controls (same antibody class and fluorophore ordered from BioLegend for CD206 and CD163 and from Miltenyi Biotec for CD80) were used to rule out unspecific binding of antibodies (Appendix A). All of the stains were performed at 4 °C for 20 min in the dark. Data were acquired using a Cytoflex flow cytometer (Beckman Coulter, Brea, CA, USA).

### 2.11. Statistical Analysis

Statistical analysis was conducted using GraphPad Prism v.8.0.2 (GraphPad Software). Differences between FBS and hPL regarding cell number, cell area, and cell viability were analyzed within each single time point and phenotype using *t*-test for matched parametric or non-parametric data depending on data distribution. Protein array, gene expression, and flow cytometry data were analyzed by One-Way ANOVA for matched non-parametric data (Friedman’s test) followed by Dunn’s post hoc test to compare the different macrophage phenotypes (i.e., M0, M1, M2a, and M2c) within the same culture condition (i.e., hPL or FBS). Differences between M0 macrophages and M0 macrophages co-cultured with chondrocytes within the same culture condition (i.e., hPL or FBS) were analyzed by *t*-test for matched parametric or non-parametric data depending on data distribution. Differences were considered statistically significant for *p*-values below 0.05.

## 3. Results

### 3.1. Effect of hPL on Macrophage Adhesion and Morphology

Cell adhesion and morphology during culture in hPL and FBS were monitored daily, taking pictures at every medium change (Figure 2a). In the differentiation phase, after 2 and 5 days of culture, more cells were observed in hPL cultures, albeit the differences in cell number between the hPL and FBS conditions were not statistically significant. This effect was maintained at day 7 for any polarization condition, although the differences between hPL and FBS remained non-significant (Figure 2b). Concerning cell morphology, during the differentiation phase, no relevant differences were observed between the hPL- and FBS-cultured macrophages. The quantification of the cell area confirmed this observation, showing that there were no significant differences between the unpolarized and polarized macrophages cultured in hPL and FBS. By comparing the different phenotypes on day 7, it was found that the M1 macrophages cultured in FBS showed an inferior area in comparison with the other phenotypes. This feature was also observed in the M1 macrophages cultured in hPL in comparison with the other phenotypes in hPL (Figure 2b). The count of live and dead cells at day 7, after macrophage detachment and before antibody staining for flow cytometry, showed that cell viability was comparable in the hPL and FBS conditions, with the percentage of live cells being between 75% and 95% for all of the analyzed conditions (Figure 2c).

### 3.2. Analysis of Macrophage Secretory Profile

The analysis of the secretory profile of hPL- and FBS-cultured macrophages revealed a general overlap between these culture conditions, indicating that the profile of up- and downregulated markers in response to cytokine-induced polarization was similar (Figure 3). The fold increase for each cytokine vs. the M0 condition is reported in Appendix A.

The highest levels of expression of the pro-inflammatory cytokines IL1α, IL1β, IL8, and TNFα and the chemokines CCL3 and CXCL9 were found in the M1 macrophages in both culture conditions. When comparing the M1 and M0 macrophages, significant differences were found for IL1α, IL1β, CCL3, and CXCL9 (*p* < 0.05) in hPL and for IL1α, IL1β, TNFα, CCL3, and CXCL9 (*p* < 0.05) in FBS. While similar levels were observed in hPL and FBS for most of these markers, we found that the M1 macrophages in hPL showed, respectively, 3.1-fold and 5.2-fold higher values of TNFα and CCL3 in comparison with the M1 macrophages in FBS. CCL2 was more expressed by the M1 macrophages in hPL, but not in FBS. On the other hand, the soluble form of CD163 was quite stable in hPL, while it was upregulated in the M1 macrophages in FBS. IL4, IL13, and CCL18 were expressed at the highest levels by M2a macrophages in both the hPL and FBS cultures, with significant differences between the M2a and M0 macrophages (*p* < 0.05). A similar behavior was observed for CCL22, although the differences were only found to be significant when comparing the M2a and M1 macrophages. Finally, IL10, MMP7, and TIMP1 were significantly upregulated by M2c macrophages in comparison to M0 cells in both hPL and FBS, although the expression levels were different between the hPL- and FBS-cultured macrophages.

Despite the similar behavior observed for many of the analyzed markers, some discrepancies emerged between the hPL- and FBS-cultured macrophages. For instance, while IL8 was clearly upregulated in the M1 macrophages cultured in hPL, this effect was not so evident for the FBS-cultured macrophages. This was mainly due to the high levels of this specific marker observed for one of the macrophage pools when the cells were cultured in FBS, as is also visible in the wide min-to-max data distribution in the graph. Additionally, CCL2 proved to be an M1 marker in macrophages cultured in hPL, but not in FBS, despite the levels of this analyte being similar in the M1 macrophages in both culture conditions. In fact, when macrophages were cultured in FBS, the levels of CCL2 were also quite high for the other phenotypes, and even higher than those associated with the M1 phenotype in the case of M0 and M2c macrophages. Finally, CCL22, which was clearly upregulated in the M2a macrophages both in hPL and FBS, showed higher levels in the hPL-cultured M0, M1, and M2c macrophages in comparison with their FBS counterparts.

### 3.3. Transcriptional Expressions of M1, M2a, and M2c Cell Surface Markers

A gene expression analysis revealed that *CD80*, generally regarded as an M1 marker, was more expressed by the M1 macrophages in comparison to the M0 macrophages in both the hPL and FBS cultures. The expressions of *CD206* and *CD163* was respectively upregulated in the M2a and M2c macrophages regardless of the culture in hPL or FBS (Figure 4). In the basal condition, all of the analyzed markers were less expressed in the hPL-cultured M0 macrophages in comparison with their FBS counterpart. In particular, the levels of *CD206* and *CD163* in the M0 macrophages cultured in hPL were very low compared to those observed in FBS-cultured M0 macrophages.

### 3.4. Macrophage Immunophenotype

To more thoroughly describe the immunophenotype of unpolarized and polarized macrophages, both the percentage of positive cells and the Mean Fluorescence Intensity (MFI) for each marker were considered (Figure 5a). Similar to gene expression, CD80 was regarded as an M1 marker, while CD206 and CD163 were, respectively, regarded as M2a and M2c markers.

The vast majority of M0 macrophages were positive for CD80 after 7 days of culture in both hPL and FBS. The M1 polarization mainly affected the level of expression of this marker rather than the percentage of positive cells, as demonstrated by the increase in MFI, which was associated with CD80 observed in the M1 macrophages in comparison with the M0 macrophages in both culture conditions. Among the hPL-cultured cells, the M1 macrophages expressed the highest percentage of CD80^+^ cells, with significant differences compared to the M2a and M2c macrophages. These data were confirmed by the analysis of the CD80-associated MFI, albeit with a high variability among macrophage pools. Among the FBS-cultured macrophages, for all phenotypes, the percentage of CD80^+^ cells ranged between 75% and 100%, with the highest expression detected in the M1 macrophages. The differences among phenotypes were more evident in terms of the CD80-associated MFI. Indeed, the CD80 MFI values were significantly higher in the M1 macrophages compared to all other phenotypes, indicating once again that, rather than depending on variations in the percentage of positive cells, the modulation of CD80 depends on variations in the expression levels.

The behavior of CD206 was very similar in the hPL- and FBS-cultured macrophages. In both conditions, the M2a macrophages displayed the highest percentage of CD206^+^ cells and the highest MFI values, with significant differences compared to the M1 macrophages, which showed the lowest values for these parameters. On the other hand, CD163 was differently regulated in hPL- and FBS-cultured cells. In hPL, the highest levels of this marker were observed for the M2c macrophages both in terms of the percentage of positive cells and MFI values. In FBS, all phenotypes displayed about 100% CD163^+^ cells, while differences were found in terms of the MFI, with the highest values observed in the M2c macrophages. Of note, the CD163-associated MFI was about 20-fold lower in the hPL-cultured macrophages than in the FBS-cultured macrophages, indicating that the expression of this M2c marker was significantly downregulated in hPL.

These results and the relationship between the percentage of positive cells (described by the bubble position along the y-axis) and MFI (described by the bubble area) for each marker are presented in the bubble graphs (Figure 5b). Overall, a reduced percentage of positive cells and a lower MFI were observed for CD206 and CD163 in the hPL-cultured macrophages. As mentioned above, the difference between the hPL- and FBS-culture macrophages was particularly relevant for CD163. In line with this evidence, the hPL macrophages were characterized by a very low percentage of CD206^+^ CD163^+^ cells, which was different from what was seen in the FBS-cultured macrophages. These were indeed characterized by strong positivity for these M2 markers, especially when considering the M0, M2a, and M2c macrophages. Representative overlay histograms for phenotype-specific markers in unpolarized and polarized macrophages are shown in Appendix A.

### 3.5. Validation of the Macrophage Co-Culture Model

The ability of articular chondrocytes expanded either in hPL or in FBS to modulate the macrophage immunophenotype was then tested. All of the experimental phases were conducted using either 10% hPL or 10% FBS as a culture medium supplement both for chondrocytes and for macrophages.

A flow cytometry analysis (Figure 6a) showed that the M0 macrophages cultured in hPL reacted to the co-culture, displaying an increase in the percentage of CD80^+^ cells. This effect was also mirrored by the increase in the CD80-associated MFI in the co-cultured macrophages in hPL. Interestingly, a similar effect was induced by the treatment with dexamethasone applied to hPL-cultured M0 macrophages (Appendix A). Differently, the expression of CD80 in the M0 macrophages cultured in FBS was not modified by the co-culture with chondrocytes. Regarding the M2a marker, the M0 macrophages co-cultured with chondrocytes showed a significant upregulation of CD206 expression in terms of the percentage of CD206^+^ cells and the MFI in both the hPL and FBS culture conditions. Finally, the expression of CD163 in the co-cultured macrophages remained similar to that of the M0 macrophages alone in both the hPL- and FBS-based culture conditions. In line with the results shown in Figure 5, the expressions of CD206 and CD163 were lower in the hPL conditions than in the FBS conditions. These data are summarized in the bubble graphs (Figure 6b), showing the relationship between the number of positive cells and the MFI for each marker. Representative overlay histograms for phenotype-specific markers in unpolarized and polarized macrophages are shown in Appendix A.

## 4. Discussion

In the present study, the behavior of unpolarized and polarized macrophages in response to the culture with human platelet lysate (hPL), a GMP-compliant alternative supplement for cell culture intended for clinical applications, was characterized. This study started from the need to assess the reaction of macrophages to this human-derived supplement in potential co-culture models aimed at determining the immunomodulatory function of candidate cells for biological therapies.

Overall, the results of this study indicate that using hPL is a suitable option for the culture of primary human macrophages and that the macrophage response to standard polarization protocols (stimulation with IFNγ and LPS for M1, with IL4 and IL13 for M2a, and with IL10 for M2c) in hPL culture conditions is consistent with that observed in FBS, although the cell surface expressions of CD206 and CD163, respectively, for M2a and M2c macrophages, is decreased by hPL.

hPL seemed to promote macrophage adhesion in comparison to FBS regardless of the polarization phenotype, which was in line with previously published data [23], albeit the differences were not statistically significant. The analysis of secreted factors, including cytokines, chemokines, and enzymes involved in tissue remodeling, allowed us to investigate the response of hPL-cultured macrophages. Proinflammatory cytokines, such as IL1β and TNFα, and chemokines were consistently upregulated in the hPL- and FBS-cultured M1 macrophages compared to their M0 counterparts, which was in line with the data in the literature [23]. Of note, CCL2 and CCL3, also known as Monocyte Chemoattractant Protein-1 (MCP-1) and Macrophage Inflammatory Protein-1α (MIP-1α), play a crucial role in the recruitment of monocytes and inflammatory cells in vivo and in the maintenance of an active immune response. It was also observed that the macrophages cultured in hPL were able to polarize into an M2a phenotype, as shown by the increased secretion of well-recognized M2a markers [25,33], such as IL4, IL13, CCL18, and CCL22, in a similar way to the FBS-cultured cells. Finally, regarding the secretory profile of the M2c phenotype, a clear induction was observed only for IL10 both in the hPL- and FBS-cultured macrophages, while a slight upregulation was observed for MMP7, which was in line with the data in the literature on FBS cultures [34,35]. To the best of our knowledge, these findings represent the only available data regarding the secretory profile of macrophages polarized in hPL culture conditions, providing valuable insight into the secretory function of hPL-cultured macrophages to other researchers that are approaching this new culture strategy with or without direct clinical purposes. Indeed, the current data in the literature are only available for a limited panel of cytokines (i.e., IL1β, IL6, IL8, and IL10) and only for M0 macrophages [23].

The ability to polarize in hPL culture conditions was confirmed by the analysis of cell surface markers by flow cytometry. As phenotypic markers, we selected CD80, CD206, and CD163, which are widely used to monitor the macrophage polarization state [25,26,27,28,29]. CD80 is involved in the activation of the immune response to pathogens, and it is associated with the M1 phenotype. CD206 is associated with M2a macrophages and is usually involved in wound healing and immune modulation. Finally, CD163 is commonly associated with M2c macrophages, which exert anti-inflammatory and tissue repair functions. The preliminary experiments conducted by our group highlighted surprisingly lower levels of CD206 and almost undetectable levels of CD163 in the M2a and M2c macrophages cultured in hPL, prompting us to investigate this specific aspect. The analysis of the literature revealed that there were no available data regarding the protein expressions of CD206 and CD163 in the macrophages cultured in hPL. Tylek and coworkers reported gene expression data for CD206 and CD163, while flow cytometry data were provided only for CD206 but without comparing hPL- and FBS-cultured macrophages [23]. Beyond this study, only one other study used hPL as a supplement for the culture of human macrophages [24]. hPL was not used to substitute FBS, but it was used in combination with FBS for a short period of time (24–48 h) [24], significantly complicating any comparison with our findings.

The results reported in the present study confirmed our preliminary findings. Overall, lower levels of typical M2a and M2c markers, namely CD206 and CD163, were detected in the macrophages cultured in hPL in response to M2-polarizing cytokines compared to their FBS counterparts. This effect might depend, at least in part, on the presence of CD40L in the hPL [36]. Indeed, the activation of the CD40–CD40L signaling axis has been reported to significantly reduce the surface expression of these markers in M2 macrophages to M1-like levels [23]. Our data also show an evident inhibitory effect of CD40L in basal conditions and might explain why, despite the increase in gene expression observed for CD206 and CD163 in the M2a and M2c macrophages, respectively, these markers were detected at low levels by flow cytometry in the hPL-cultured macrophages. However, an additional investigation is necessary to confirm whether CD40L plays a role in downregulating these markers in our experimental set-up. Finally, functional co-culture experiments were performed to evaluate the ability of this in vitro macrophage model to verify the immunomodulatory potential of specific cells that could be exploited in pathologies with an inflammatory component. For this purpose, the immunomodulatory potential of chondrocytes was tested in view of their use to treat cartilage degenerative lesions. This experimental co-culture set-up was already used in our laboratories to assess the immunomodulatory function of adipose-derived stem cells (ASCs), using FBS as a medium supplement [37]. The same cell concentrations and unpolarized macrophages were selected to assess the immunomodulatory ability of chondrocytes in the presence of either hPL or FBS. Flow cytometry was selected as the readout technique, as it is a quantitative and rapid assay which can be easily implemented as a routine quality control method to determine the immunomodulatory function of candidate cells for clinical applications.

Our findings show that the macrophages reacted to the co-culture with chondrocytes by increasing the expression of the M2a marker CD206 in both culture conditions, suggesting that there was an active interaction between macrophages and chondrocytes. On the other hand, the expression of the M2c marker CD163 showed no changes in the macrophages co-cultured with chondrocytes. The results of the co-culture experiments confirm that the expressions of CD206 and CD163 were lower in the hPL conditions than in the FBS conditions.

As a limitation of this study, we would like to stress that our findings should be considered as being limited to the specific polarization protocols applied here, which, despite being widely used, do not represent the only available strategies to induce primary human macrophages into specific phenotypes. For instance, using dexamethasone instead of M2 polarizing cytokines might yield partially different outcomes in terms of cell surface marker expression. Similarly, differences in the hPL production process might lead to discrepancies in its biological activity, as reported for the culture of mesenchymal stem cells [38]. The hPL used in this study was leukodepleted through filtration, subjected to platelet fragmentation by three freeze–thaw cycles, and centrifuged to remove platelet bodies before storage. In case a different production protocol is used to produce hPL, we strongly recommend proceeding with macrophage characterization before setting up any co-culture model. From a wider point of view, considering the use of this co-culture model to assess the immunomodulatory function of cell-based therapies in the context of osteoarthritis, a limitation is the lack of synovial fibroblasts in the system. Fibroblasts reside in close contact with macrophages in the synovial membrane, and the crosstalk between these cells can also affect the interaction between articular chondrocytes and macrophages. To explore this aspect, a triculture should be established.

In summary, our results demonstrate that markers typically regarded as phenotype-specific in FBS-cultured macrophages are confirmed in hPL-cultured macrophages, indicating that this culture condition does not affect the responsiveness of macrophages to standard polarization protocols. The validation of the model with co-cultured MSCs indicates that using hPL-cultured macrophages is a valuable option to test the immunomodulatory potential of hPL-expanded cells for clinical use. However, the different basal expressions of CD206 and CD163 observed in the hPL-cultured macrophages suggest that a careful interpretation of these specific markers is needed in immunomodulatory assays.

## Figures and Tables

**Figure 1 biomedicines-12-00833-f001:**
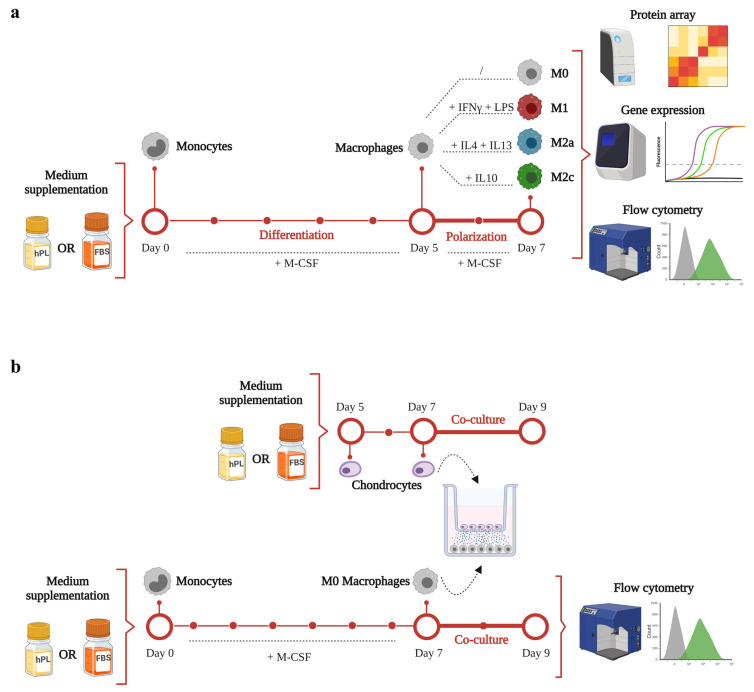
A schematic representation of the experimental design and methodology of this study. (**a**) timeline and analyses applied for macrophage differentiation and polarization in hPL or FBS. (**b**) timeline and methodology applied for the co-culture set-up. This figure was created with BioRender.com.

**Figure 2 biomedicines-12-00833-f002:**
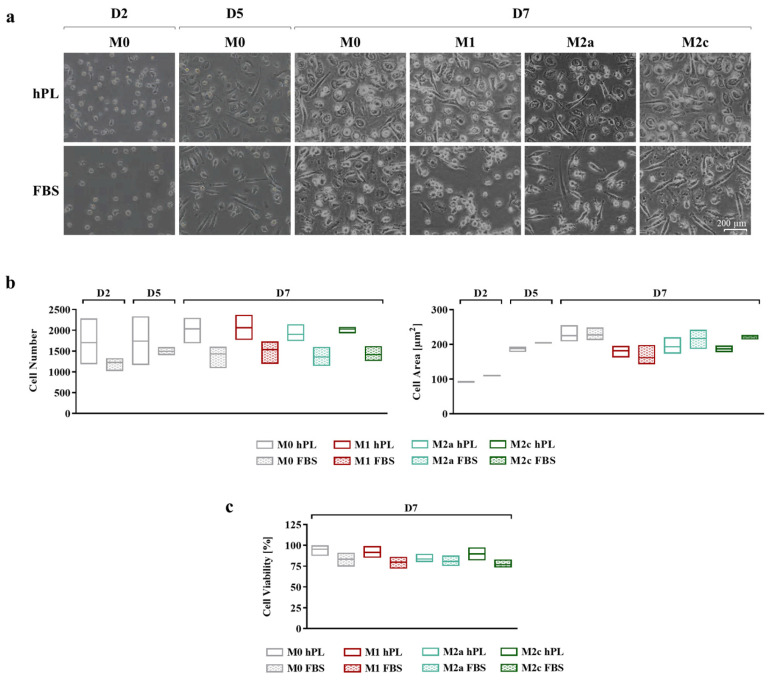
(**a**) Representative pictures showing cell morphology at different time points of human primary macrophages during differentiation and polarization in either hPL or FBS (scale bar: 200 µm). (**b**) Quantification of cell number (ROI size: 2200.60 µm × 1660.80 µm) and cell area based on image analysis from three batches of macrophages (mean ± SD). (**c**) Evaluation of macrophage viability based on count of live and dead cells and expressed as percentage of live cells over total cell number (mean ± SD).

**Figure 3 biomedicines-12-00833-f003:**
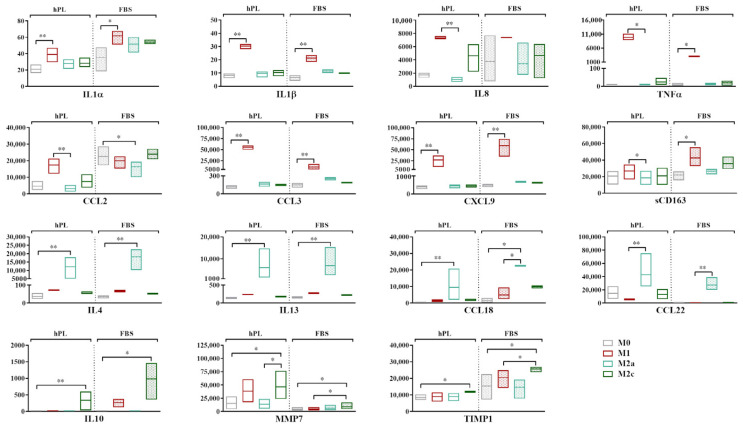
Secretory profiles of unpolarized and polarized macrophages cultured in hPL or FBS. Min-to-max box plots represent concentration of each marker expressed as pg/mL. Data were obtained from independent experiments conducted with three different batches of macrophages (* *p* < 0.05, ** *p* < 0.01). Values of CCL18 in M2a macrophages in FBS fell above range of assay and are indicated as Out of Range Above (>OOR).

**Figure 4 biomedicines-12-00833-f004:**
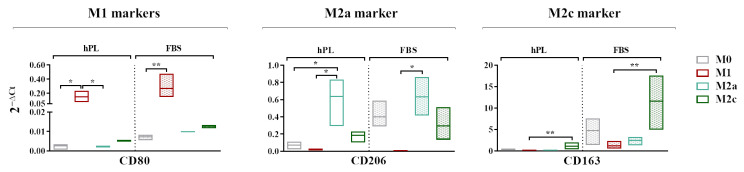
A gene expression analysis showing the transcriptional levels at day 7 of phenotype-specific cell surface markers measured in unpolarized and polarized macrophages cultured in hPL or FBS. The min-to-max box plots show data obtained from independent experiments conducted with three different batches of macrophages (* *p* < 0.05; ** *p* < 0.01). The data are normalized on the housekeeping gene (2^−ΔCt^ method).

**Figure 5 biomedicines-12-00833-f005:**
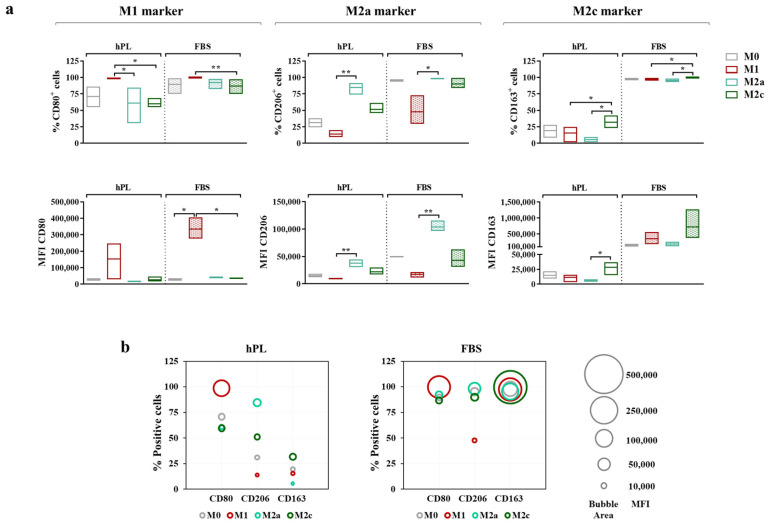
Cell surface marker expressions in unpolarized and polarized macrophages cultured in hPL or FBS. (**a**) The min-to-max box plots (with line at mean) show data expressed both as the percentage of positive cells and as the Mean Fluorescence Intensity (MFI) of the cell population (* *p* < 0.05; ** *p* < 0.01). (**b**) The bubble graphs show the relationship between the number of positive cells and the MFI, where the position of the bubble along the y-axis indicates the percentage of positive cells and the area of the bubble is proportional to the MFI. The data were obtained from independent experiments conducted with three different pools of macrophages.

**Figure 6 biomedicines-12-00833-f006:**
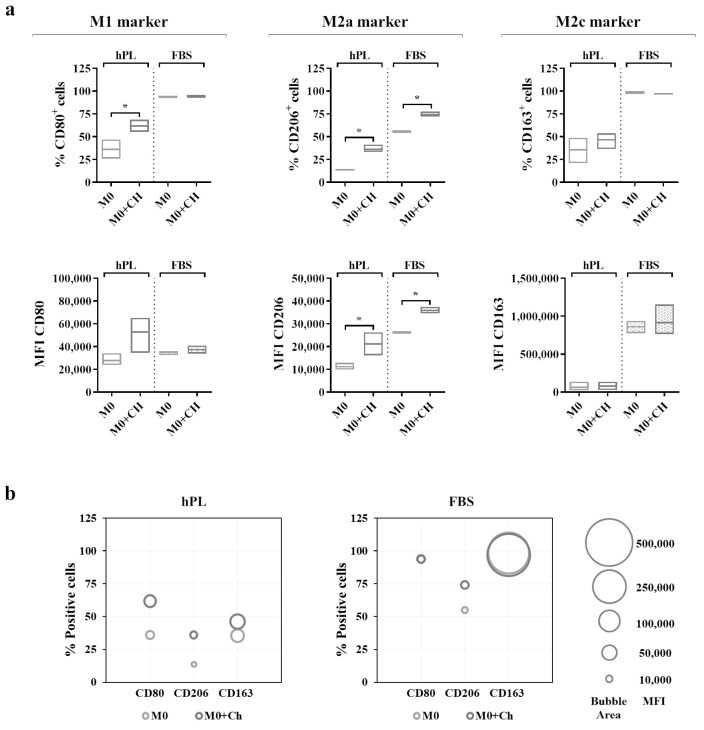
(**a**) Cell surface marker expressions in unpolarized macrophages cultured in hPL to compare the phenotype of macrophages alone (M0) with that of macrophages co-cultured with chondrocytes (M0+Ch). The min-to-max box plots (with line at mean) show data expressed both as the percentage of positive cells and as the Mean Fluorescence Intensity (MFI) of the cell population (* *p* < 0.05). (**b**) The bubble graphs show the relationship between the number of positive cells and MFI, where the position of the bubble along the y-axis indicates the percentage of positive cells, and the area of the bubble is proportional to the MFI. The data were obtained from independent experiments conducted with 3 different pools of macrophages.

## Data Availability

The data presented in this manuscript are available at https://osf.io/ph293/ (DOI 10.17605/OSF.IO/PH293).

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
