# Peer review of "Using Macrophage Polarization in Human Platelet Lysate to Test the Immunomodulatory Potential of Cells for Clinical Use"

_biomedicines, 2024, doi:10.3390/biomedicines12040833_

Round 1

Reviewer 1 Report

Comments and Suggestions for Authors

The manuscript describes experiments demonstrating potential usefulness/limitations of human platelet lysate which is often used as a replacement of serum in culture media. Authors set several experiments to compare serum-supplemented and hPL-supplemented cultures containing differentiating macrophages alone or in the presence of human chondrocytes in trans-well settings. The results of pretty much technical study are quite sound and mostly support the conclusions that hPL has some limitations as serum replacement. The article is well-written and contains nicely presented data that are easy for understanding. Despite overall quality and soundness of the study I have several experimental issues that have to be responded/corrected by the authors prior to acceptance:

1) Figure 2a shows cell morphology and ability to spread on the surface of plastic. I am not sure that these parameters can appropriately measure the proportion of live cells. Please, use flow cytometry of cells stained with vital dye to show the comparable proportions/absolute numbers of live cells in the cultures;

2) In co-culture experiments the authors chose chondrocyte to characterize effects of cell-derived molecules on the maturation of monocyte-derived macrophages in hPL-suppleneted cultures. Will this effect reproduce if one would add fibroblasts instead of chondrocytes? Did you try cells of other origin which can be found in inflamed tissues? Please, comment, or provide extra data.

3) It is surprizing that in experiment with chondrocytes the authors did not use for control media supplemented with FBS. I insist that this control is needed to improve the quality and value of the paper, since suggested comparison of FBS and hPL is logical and will complement the data obtained in the first experiment with macrophage polarization.

Concluding, I have to ask for major revision, responce to the raised issues and addition of missing data.

Author Response

Reviewer 1

Comments and Suggestions for Authors

The manuscript describes experiments demonstrating potential usefulness/limitations of human platelet lysate which is often used as a replacement of serum in culture media. Authors set several experiments to compare serum-supplemented and hPL-supplemented cultures containing differentiating macrophages alone or in the presence of human chondrocytes in trans-well settings. The results of pretty much technical study are quite sound and mostly support the conclusions that hPL has some limitations as serum replacement. The article is well-written and contains nicely presented data that are easy for understanding. Despite overall quality and soundness of the study I have several experimental issues that have to be responded/corrected by the authors prior to acceptance:

1) Figure 2a shows cell morphology and ability to spread on the surface of plastic. I am not sure that these parameters can appropriately measure the proportion of live cells. Please, use flow cytometry of cells stained with vital dye to show the comparable proportions/absolute numbers of live cells in the cultures;

The authors thank Reviewer for appreciating the manuscript. Regarding the assessment of macrophage proliferation, at day 7 we performed trypan blue viable counts of cells after detachment and before making antibody staining for flow cytometry analysis. The cell viability data expressed as % of live cells for all the phenotypes cultured either in hPL or in FBS are now shown in Figure 2, panel c and Results were implemented as follows:

“The count of live and dead cells at day 7, after macrophage detachment and before antibody staining for flow cytometry, showed that cell viability was comparable in hPL and FBS conditions, with the percentage of live cells being comprised between 75% and 95% for all the analyzed conditions (Figure 2c).”

2) In co-culture experiments the authors chose chondrocyte to characterize effects of cell-derived molecules on the maturation of monocyte-derived macrophages in hPL-supplemented cultures. Will this effect reproduce if one would add fibroblasts instead of chondrocytes? Did you try cells of other origin which can be found in inflamed tissues? Please, comment, or provide extra data.

In this specific study, we focused on the immunomodulatory ability of human chondrocytes intended as the ability of these cells to modulate macrophage phenotype. Indeed, we are conducting a big project aimed to evaluate the use of chondrocytes to treat cartilage degenerative lesions within osteoarthritic joints. Being osteoarthritic joints characterized by an inflammatory milieu, we were interested in assessing the ability of these cells not only in regenerating the cartilage defect(s), but also in restoring a more physiological microenvironment. This is the only cell type of articular origin that we have tested so far with this specific aim and at the moment we cannot speculate whether fibroblasts cultured in the same conditions would exert a similar effect on human macrophages. However, to acknowledge the possible role of fibroblasts in the context of OA research, we have added the following paragraph to the Discussion:

“From a wider point-of-view, and considering the use of this co-culture model to assess the immunomodulatory function of cell-based therapies in the context of osteoarthritis, a limitation is the lack of synovial fibroblasts in the system. Fibroblasts reside in close contact with macrophages in the synovial membrane and the crosstalk between these cells can affect also the interaction between articular chondrocytes and macro-phages. To explore this aspect a triculture should be established.”

3) It is surprizing that in experiment with chondrocytes the authors did not use for control media supplemented with FBS. I insist that this control is needed to improve the quality and value of the paper, since suggested comparison of FBS and hPL is logical and will complement the data obtained in the first experiment with macrophage polarization.

As requested by the Reviewer we added the data concerning the co-culture of macrophages and chondrocytes in FBS in Figure 6. Following these changes, the section 3.5. “Validation of the macrophage co-culture model” has been extensively revised to include the new findings. In these experiments, that were conducted before the experiments in hPL, we did not use dexamethasone as control group. This is the reason why we initially did not include the FBS data in the manuscript. In the revised version of the manuscript, to be consistent, we removed the dexamethasone group in hPL from Figure 6. The data concerning M0 macrophages in hPL cultured with dexamethasone are now included in Supplementary Figure A4.

Concluding, I have to ask for major revision, responce to the raised issues and addition of missing data.

Reviewer 2 Report

Comments and Suggestions for Authors

The aim of this study was to develop a macrophage polarization model in human platelet lysate (hPL), evaluating both the secretory profile and the expression of specific markers for pro-inflammatory or anti-inflammatory phenotypes. The authors conducted a comparative analysis of macrophage behavior in hPL and fetal bovine serum (FBS) cultures to determine if the specific phenotypes exhibited similar marker expression in both culture conditions. Additionally, the experimental validity of the model was confirmed through a functional assay involving co-culture of macrophages and chondrocytes, mimicking the intercellular communication within a joint microenvironment. This was particularly relevant in the context of using chondrocytes for cell-based therapy in osteoarthritis (OA) patients.

The results indicated comparable marker regulation between hPL and FBS cultures, although hPL-cultured macrophages exhibited lower basal levels of CD206 and CD163. Furthermore, functional co-culture experiments with chondrocytes demonstrated increased CD206 expression, suggesting an interaction between macrophages and chondrocytes.

Overall, the study's objectives are clearly defined, the experimental procedures are well-executed, and the methods are meticulously described. The findings are robust, and the discussion provides comprehensive insights into the implications of the results.

I have only few comments to this excellent work.

Introduction: I propose reorganizing the introduction to begin with a concise overview of the pathophysiology of osteoarthritis, elucidating the rationale and methodologies behind the study of cellular models of osteoarthritis, with a particular focus on the role of macrophages (essentially previewing the paragraphs between lines 67 and 75). Additionally, I would delve into the significance of co-cultures and offer other examples of co-culture models utilized for osteoarthritis (i.e. Favero M et al. J Cell Physiol. 2019;234(7):11176-11187. doi:10.1002/jcp.27766), while also elucidating the mechanism of cross-talk. After this introductory overview, I would proceed to relocate the initial two paragraphs concerning FBS and hPL. 

Line 75-77: please specify the difference between M2a and M2C macrophages.

Author Response

Reviewer 2

Comments and Suggestions for Authors

The aim of this study was to develop a macrophage polarization model in human platelet lysate (hPL), evaluating both the secretory profile and the expression of specific markers for pro-inflammatory or anti-inflammatory phenotypes. The authors conducted a comparative analysis of macrophage behavior in hPL and fetal bovine serum (FBS) cultures to determine if the specific phenotypes exhibited similar marker expression in both culture conditions. Additionally, the experimental validity of the model was confirmed through a functional assay involving co-culture of macrophages and chondrocytes, mimicking the intercellular communication within a joint microenvironment. This was particularly relevant in the context of using chondrocytes for cell-based therapy in osteoarthritis (OA) patients.

The results indicated comparable marker regulation between hPL and FBS cultures, although hPL-cultured macrophages exhibited lower basal levels of CD206 and CD163. Furthermore, functional co-culture experiments with chondrocytes demonstrated increased CD206 expression, suggesting an interaction between macrophages and chondrocytes.

Overall, the study's objectives are clearly defined, the experimental procedures are well-executed, and the methods are meticulously described. The findings are robust, and the discussion provides comprehensive insights into the implications of the results.

I have only few comments to this excellent work.

The authors thank the Reviewer for appreciating the manuscript.

Introduction: I propose reorganizing the introduction to begin with a concise overview of the pathophysiology of osteoarthritis, elucidating the rationale and methodologies behind the study of cellular models of osteoarthritis, with a particular focus on the role of macrophages (essentially previewing the paragraphs between lines 67 and 75). Additionally, I would delve into the significance of co-cultures and offer other examples of co-culture models utilized for osteoarthritis (i.e. Favero M et al. J Cell Physiol. 2019;234(7):11176-11187. doi:10.1002/jcp.27766), while also elucidating the mechanism of cross-talk. After this introductory overview, I would proceed to relocate the initial two paragraphs concerning FBS and hPL.

We would like to acknowledge the Reviewer for this helpful suggestion, which has greatly improved the organization of the Introduction. As suggested by the Reviewer, we have moved the paragraph regarding the OA to the beginning, specifying the role of macrophages in joint inflammation more clearly. Additionally, we have included a paragraph about the use of co-cultures in OA research, specifying the purposes of these studies and providing relevant references. Following this, we have included the paragraph concerning the use of FBS and hPL.

Line 75-77: please specify the difference between M2a and M2C macrophages.

Based on the Reviewer’s request, the following sentence with reference has been added to the Introduction:

“M2 macrophages can be further divided in different subpopulations which include M2a macrophages mainly related to anti-inflammatory activity, and M2c macrophages mainly related to tissue repair [11] ”

Round 2

Reviewer 1 Report

Comments and Suggestions for Authors

The authors have done very good job on addressing my concerns. 

I have to admit that now the manuscript is worth publishing in its current form without any delays. Congratulations! My decision is "Accept"!